# Potential Teratogenicity Effects of Metals on Avian Embryos

**DOI:** 10.3390/ijms251910662

**Published:** 2024-10-03

**Authors:** Rita Szabó, Péter Budai, Éva Juhász, László Major, József Lehel

**Affiliations:** 1Department of Plant Protection, Institute of Plant Protection, Georgikon Campus, Hungarian University of Agriculture and Life Sciences, Deák F. u. 16, H-8360 Keszthely, Hungary; budai.peter@uni-mate.hu (P.B.); major.laszlo@phd.uni-mate.hu (L.M.); 2Crop Science Division, Bayer Hungária Ltd., Dombóvári u. 26, H-1117 Budapest, Hungary; eva.juhasz@bayer.com; 3Department of Food Hygiene, Institute of Food Chain Science, University of Veterinary Medicine Budapest, István u. 2, H-1400 Budapest, Hungary; 4National Laboratory for Infectious Animal Diseases, Antimicrobial Resistance, Veterinary Public Health and Food Chain Safety, University of Veterinary Medicine Budapest, István u. 2, H-1400 Budapest, Hungary

**Keywords:** cadmium, copper, lead, developmental abnormalities, mortality, environmental safety

## Abstract

Agricultural areas can provide sources of food and hiding and nesting places for wild birds. Thus, the chemical load of potentially toxic elements (Cd, Cu, Pb) due to industrial and agricultural activities can affect not only the adult birds but also the embryos developing in the egg. The toxic effects of heavy metals applied alone were investigated on chicken embryos in the early and late stages of embryonic development using injection and immersion treatment methods. On day 3 of incubation, permanent preparations were made from the embryos to study the early development stage. There were no significant differences observed in embryo deaths and developmental abnormalities in this stage. On day 19 of incubation, the number of embryonic deaths, the body weight of the embryos, and the type of developmental abnormalities were examined. The embryonic mortality was statistically higher in the groups treated with cadmium and lead in the case of the injection treatment. A significant increase in developmental disorders was observed in the copper-treated group using the immersion application. The body weight significantly decreased in the cadmium- and lead-treated group using both treatment methods. However, a significant change in the body weight in the copper-treated group was only realized due to the injection method.

## 1. Introduction

In the last century, various metals (e.g., mercury, arsenic) were widely used in crop protection to eliminate harmful pests and microorganisms (insects, fungi). Due to the rapid development of toxicology, more and more research projects and studies have come to light that report on the pronounced toxicity of various compounds of metals. Based on the results, the licensing authorities have restricted or banned the use of many metal compounds as pesticides. However, the entire biosphere has been polluted due to the extensive use in the past [1,2,3].

Toxic heavy metals have been increasingly entering the environment since the beginning of the industrial revolutions. Even though the use of heavy metals has been banned in several industrial and agricultural processes and activities, they will be/may be the cause of the most serious environmental damage in the coming decades. Contamination of soils, groundwater, and surface water with heavy metals can cause severe environmental stress, which can lead to environmental damage and endanger the health of living beings [4,5,6].

At present, the pollution of the environment with potentially toxic elements (PTEs) has still been increasing because of anthropogenic activities, such as different industrial processing methods, mining activities, agricultural activities, geochemical changes (erosion, sedimentation, decomposition), and irregular use and disposal of waste materials [7,8,9,10].

Furthermore, PTEs can be released into surface and groundwaters and soils from natural sources (e.g., fossil fuels, volcanic rocks, ores) and can contribute to the contamination of the environment [8,11,12,13,14].

PTEs have a high capability of being accumulated in the soil and water, and, through them, in plants and birds [15]. In aquatic ecosystems, heavy metals can dissolve in the water and can be taken up by fish and can be accumulated in microscopic organisms or can bind to the sediment [16,17,18].

PTEs can enter the body of piscivorous birds by ingestion of a contaminated diet (e.g., plant, fish), through the water, and from the surrounding environment, and thus they can be absorbed and distributed throughout the body, be accumulated in different tissues of birds, or even be excreted into the feathers or eggs [19,20,21,22,23,24,25,26].

Burger [27] described that Cd and Pb (alongside selenium, manganese, and chromium) can be excreted from the body via eggs in the roseate tern (*Sterna dougallii*) and the herring gull (*Larus argentatus*), and can be detected in the eggshell [27].

Copper has a higher ability to accumulate in the mallard (*Anas platyrhynchos*) eggshell than lead, probably due to different pathways of accumulation [28].

However, the PTE content of the egg can originate from contaminated materials as direct contamination, or contaminated water can pose a potential source of metals for feathers during nestling and egg laying [29,30,31]. The metals can penetrate through the eggshell and thus can pose a potential risk to the avian embryo [32].

PTEs can be detected in the egg white and egg yolk [14,33,34,35,36,37,38], and even in the eggshell [39].

The presence of PTEs in the egg represents hazards because they can influence embryogenesis, resulting in developmental abnormalities or other malformations [40,41,42].

However, PTEs can act and react with other xenobiotics (e.g., different pesticides) released into the environment, and thus can induce toxic interactions, resulting in closely similar or even more severe effects [43,44,45,46].

### 1.1. Cadmium

Cadmium occurs in small quantities in the earth’s crust, with a concentration of 0.1–1 mg/kg [47]. Sedimentary rocks can contain 10–20 times higher amounts [48,49]. It is primarily found in zinc-containing ores, and to a lesser extent is bound to lead and copper [50].

It can be measured at a concentration of <1 µg/L in surface and groundwaters, and 0.001–0.1 µg/mL in oceans, but is present in larger quantities in areas rich in phytoplankton [48,49].

Cadmium accumulates in plants, which is enhanced by phosphate-containing fertilizers, acidic soils, and industrial wastewater used for irrigation [51]. The consumption of these plants can induce expressed exposure to the wild animals [52].

Cadmium (Cd) can enter the body of animals through contaminants in the air, drinking water, and feed, and, because of repeated exposure, Cd can accumulate in various organs, mainly in the liver and kidneys [53,54,55]. Based on the literature, the maximum tolerable dietary level of Cd in domestic animals is 0.5 mg/kg, tenfold of that amount (5 mg/kg) already has a toxic effect [56].

Cadmium has a toxic effect on the testes, altering the development and morphology of the testes, and can inhibit spermatogenesis [57].

Cd can increase egg production at a dose of 3 mg/kg feed. However, a higher amount (12 mg/kg feed) can reduce it and that of the thickness of the eggshell [58]. Egg production also decreased in chickens and Japanese quails by feeding on cadmium at a dose of 50 and 75 mg/kg of feed, respectively [59,60,61,62], and the number of fertile eggs was reduced [63]. Egg production and feed intake decreased because of high-dose oral cadmium exposure in poultry, and stress sensitivity increased [20]. Cd can cause morphological alterations in the ovaries of hens [64]. It has potential hepatotoxic [64], nephrotoxic [65], teratogenic [66,67], mutagenic [68], and carcinogenic effects [69,70] because it can bind directly to DNA and can inhibit the synthesis of cytochrome P450 enzymes [71,72]. Furthermore, it has an undesirable effect on the reproductive system and spermicide property [73]. The cytotoxic effect of Cd can lead to apoptosis at a lower dose, while a higher dose can induce necrosis [74,75].

### 1.2. Copper

Copper occurs on average in the earth’s crust at an amount of 70 mg/kg and in seawater at an amount of 0.003 mg/kg [76]. Today, copper is mainly used for the production of wire, as a catalyst in the chemical industry, and for the production of alloys. It is used as a nutrient in crop cultivation and can be measured in plants at a concentration of 14 mg/kg. It is already toxic to lower organisms in small concentrations, so it is used as a plant protection agent [76,77]. Copper-containing compounds are used in animal husbandry due to their feed supplement and growth-stimulating and disease-preventing effects. Its average concentration in animals is 2.4 mg/kg [76,78,79]. It is an essential trace element that is essential for the formation of hemoglobin and the smooth functioning of many enzymes [80,81].

Copper (Cu) is an essential micromineral for all animals, plays a part in many cellular metabolic processes and is a component of enzymes such as cytochrome c oxidase, Cu–Zn superoxide dismutase, tyrosinase, and lysyl oxidase [82]. However, its intake in doses significantly exceeds the level that causes poisoning. Due to the toxic effect of copper, liver failure and hemolytic anemia are developed, but damage also occurs in the kidneys, brain, and other organs [83]. The results of copper toxicity studies have shown that the induction of ROS plays a key role in the damaging effect of copper at the molecular level [84].

Copper sulfate and oxide can reduce the egg production in laying hens if it is applied at a dose of 400 mg/kg in the feed [85,86].

Furthermore, the hatchability of eggs is also reduced, and abnormal eggs can be observed (shell-less, abnormal shape and texture) in domestic bird species such as laying hens and broiler breeder hens [87,88,89].

Bhunya and Jena [90] described the genotoxic potential of copper sulfate in chicks in vivo using a chromosome aberration and micronucleus test, showing that it can induce developmental aberrations.

### 1.3. Lead

Lead in the earth’s crust is approximately present at an amount of 13 mg/kg, but this varies depending on the area and the type of soil, e.g., 10–20 mg/kg in volcanic and sedimentary rocks, 10–70 mg/kg in sandstone and carbonaceous material shale, and 100 mg/kg in phosphate-containing rock. The natural occurrence of lead is relatively small; however, the pollution released into nature through human activities (smelters, foundries, chemical production, battery production) is more significant [48,91].

Among metals, lead (Pb) is one of the most toxic and, based on current knowledge, it has no physiological function in animals. Depending on the exposure parameters, the adverse effects of Pb can range from mild biochemical or physiological alterations to disorders with severe organ dysfunction, which can occur in the liver, kidney, heart, brain, testes, and hematopoietic tissues. Death due to plumbism remains a common cause of death in both domestic and wild animals, especially birds [92,93]. Lead absorbed from the alimentary tract and lungs is concentrated in soft tissues such as the liver and kidneys during primary distribution, and later, due to repeated distribution, it accumulates in bones, where it is present for the rest of the animal’s life [94].

Lead acetate can induce the reduction of egg production in Japanese quails (*Coturnix coturnix japonica*) at a dose of 1 mg/kg feed, and in laying hens with an amount of 10–100 mg/kg feed [95,96].

Pb, which can disturb the function of the reproductive system, results in the reduced motility of sperm or even infertility in males, and can induce developmental abnormalities [97,98].

The weight of the testes significantly reduced due to the Pb treatment in ringed turtle doves (*Streptopelia risoria*), and the significant degeneration observed during the histological examination of the testes may indicate a reduced male reproductive capacity [99].

In red-legged partridges (*Alectoris rufa*) treated with three lead shot pellets, the hatching rate of females and the motility of male sperm decreased. At the same time, in birds treated with one lead shot pellet, the weight of the eggs of the females and the vigor of the male sperms increased [100].

## 2. Results

### 2.1. Early Developmental Stage

#### 2.1.1. Embryo Mortality

Injection method

During processing, there were no dead embryos in the control group on day 2 and 3 after injection. However, the embryo mortality increased to 10.00% (1/10; *p* = 0.5) in the group treated with copper sulfate and lead acetate, and to 20.00% (2/10; *p* = 0.2368) in the cadmium-treated group. Based on the status of the embryos, the majority of them died on day 1 of incubation in copper sulfate (100.00%) and on day 2 of hatching in the lead acetate-treated group (100); however, the mortality rate was the same on day 1 and 2 of hatching in the group treated with cadmium sulfate.

Immersion method

Embryo mortality was not detected in the groups of control, copper, and cadmium sulfate, but it was elevated to 10.00% (1/10; *p* = 0.5) in the lead acetate-treated group, and the embryos died on day 1 of hatching.

#### 2.1.2. Developmental Aberration

Injection method

A retarded vascular system was manifested in two embryos in the control group (20.00%). Reduced growth and underdevelopment of the vascular network appeared in one embryo (11.11%; *p* = 0.8761) due to copper sulfate. In the group treated with cadmium sulfate, teratologic malformations were observed in two embryos (25.00%; *p* = 0.6176), resulting in nine somites fused together at the head, distorted Siamese twins, and abnormal differentiation of the brain and eyeballs. Developmental abnormality was not detected in the embryos treated with lead acetate (0.00%; *p* = 1).

Immersion method

A blood ring was found in one embryo (10.00%) from the control group. There were no malformations due to copper or cadmium sulfate; and lead acetate (0.00%; *p* = 1).

### 2.2. Late Developmental Stage (on Day 19)

#### 2.2.1. Embryo Mortality

Injection method

There were four dead embryos (8.00%) in the control group on day 19 of hatching. Embryo mortality increased to 12.00% in the group treated with copper sulfate without significant difference (*p* = 0.3703). However, the increase in embryo mortality was significant in the cadmium-injected group (22.00%; *p* = 0.0452) and in the lead-treated eggs (36.00%; *p* = 0.0006).

Immersion method

Four dead embryos (8.00%) were found in the control group. Elevated embryo mortality was detected at 2.00% (*p* = 0.9718), 10.00% (*p* = 0.5), and 12.00% (*p* = 0.3703) due to treatment with copper sulfate, cadmium sulfate, and lead acetate, respectively.

#### 2.2.2. Developmental Aberration

Injection method

An opened abdominal cavity was manifested in one embryo (2.17%) from the control group. A significant increase in abnormalities (13.64%; *p* = 0.0485) was observed, including edema of the head and the neck, and curvature of the axis of the toes due to treatment with copper sulfate. A lower ratio of deformities was detected in the cadmium-treated group (10.26%; *p* = 0.1328), and edema developed on the scalp, along with curvature of the neck axis, abnormal position of the leg, and shortening of the upper beak. The ratio of the deformities was 9.38% (*p* = 0.1851) due to lead, including opened thoracic cavity, edema developed on the scalp, and curvature of the neck axis.

Immersion method

There were no abnormalities in the control group or due to lead acetate (0.00%). The developmental aberrations significantly (12.24%; *p* = 0.0160) increased in the group treated with copper sulfate, and edema of the head and the neck, curved neck, shortening of the upper beak, and curvature of the axis of the toes were manifested. A significant ratio of anomalies (8.88%; *p* = 0.0557) was detected in the cadmium-treated group, including edema developed at the neck and abnormal position of the leg.

Comparing the two treatment methods, in terms of mortality, no statistically significant difference could be proven in the early stages of development (Cu: *p* = 0.5; Cd: *p* = 0.2368; Pb: *p* = 0.7631). In the late stage of development, a significant difference was observed in the case of lead treatment methods (*p* = 0.0045), but it was not realized in the case of Cu (*p* = 0.0558) or Cd (*p* = 0.0856). In the case of developmental disorders, neither in the early (Cu: *p* = 0.4736; Cd: *p* = 0.1830; Pb: *p* = 1) nor in the late developmental stage (Cu: *p* = 0.5418; Cd: *p* = 0.5598; Pb: *p* = 0.0705) could any difference be verified statistically between the treatment methods.

#### 2.2.3. Body Weight

Injection method

The body weight of the embryos was 26.48 ± 2.01 g in the control group. Due to the treatment with copper sulfate, cadmium sulfate, and lead acetate, the body weights were significantly increased to 24.65 ± 2.37 g (*p* = 0.0021), 24.21 ± 2.38 g (*p* = 0.0003), and 24.19 ± 2.35 g (*p* = 0.0004), respectively (Figure 1).

Immersion method

The body weight of the embryos was 25.42 ± 2.12 g in the control group. Its significant decrease was detected in the groups treated with cadmium sulfate (23.72 ± 2.36 g; *p* = 0.0025) and lead acetate (23.69 ± 2.32 g; *p* = 0.0013), respectively. However, there was no significant difference in the case of copper sulfate (24.51 ± 2.29; *p* = 0.3345) compared with the control (Figure 2).

#### 2.2.4. Skeleton Staining

A skeleton-stained control embryo without deformities on day 19 of hatching is presented in Figure 3.

Injection method

Developmental disorders (retarded growth, curved neck) were detected in two embryos (20.00%, 2/10) in the control group after skeleton staining.

Growth retardation and abnormal leg position were manifested in three embryos (30.00%, 3/10) due to treatment of copper sulfate. In cadmium-treated embryos, abnormal position of the neck and growth retardation were observed in two embryos (20.00%, 2/10). A similar ratio of abnormalities (20.00%, 2/10) was realized in the embryos treated with lead acetate, resulting in abnormal leg position and growth retardation.

Immersion method

A curved neck was detected in one embryo from the control group (10.00%, 1/10). A similar malformation (slight curvature of the neck) and ratio (10.00%, 1/10) was seen in the group treated with cadmium sulfate. The development disorders increased to 20.00% (2/10) due to copper sulfate and lead acetate, and abnormal leg position and slight curvature of the neck were manifested.

#### 2.2.5. Histopathology

The structures of the neck muscle and liver (Figure 4) were physiologic and species specific. There were no histopathological lesions caused by the injection or the immersion method in the control group or the groups treated with copper sulfate, cadmium sulfate, and lead acetate.

## 3. Discussion

### 3.1. Mortality

The mortality rate was 33.30% and 16.00% after treatment with Pb by the injection and immersion methods and was 57.40% and 43.70% due to the application of Cd by both techniques in mallard eggs (*Anas platyrhynchos*) [101]. It was similar in the chicken embryos of our study at a late developmental stage due to the injection of copper (12.00%), cadmium (22.00%), and lead (36.00%), but it was lower in the case of the immersion method (Cu: 20.00%, Cd: 10.00%, Pb: 12.00%). However, the embryo mortality was lower (0–20.00%) in the chickens in our study in the case of all the investigated metals (Cd, Cu, Pb) applied by both techniques at an early developmental stage, but it was higher in embryos treated with Cd (injection: 20.00%; immersion: 10.00%).

Similarly, the increase in embryo mortality caused by Cd was also found in laboratory mammals [102].

Kertész et al. described that copper does not influence the embryonic mortality in the mallard (*A. platyrhynchos*) after the immersion treatment, but is significantly influenced by Pb, probably due to the differences in the applied dose, time, and duration of exposure [28].

Cadmium, by binding to the thiol groups of membrane proteins, causes their depolarization in the mitochondria, which leads to ATP deficiency and necrotic cell death. In other cases, depending on the cell type, it induces the efflux of mitochondrial enzymes, which in turn causes apoptotic cell death [49]. After all, these changes can lead to the death of the embryonic disc and the embryo at all stages.

Similarly, the lead most likely binds to proteins in the body, and changes their function, inhibits or mimics the effect of calcium, can replace zinc in various enzymes, and causes oxidative stress. By binding to sulfhydryl, amine, phosphate, and carboxyl groups, it modifies the binding capacity and enzyme activity of proteins [103,104,105,106]. These effects can lead to abnormal function of enzymes and biochemical processes and can result in deformity of the organs, or can even lead to the death of the embryo.

Elemental copper is not toxic; of copper salts, 400–900 mg/kg of copper sulfate is the single toxic dose for poultry. Copper salts are only slowly and partially absorbed from the stomach and intestines [107]. After absorption, copper binds to albumin and amino acids in the blood. The circulating copper ions form hemocuprein and ceruloplasmin with blood proteins, erythrocuprein in red blood cells, and are stored in the liver in the form of hepatocuprein and bound to metallothionein [108]. The primary organ of copper storage and metabolism within the body is the liver, where it accumulates primarily in the nuclei, mitochondria, and lysosomes [109]. The liver is usually able to store and detoxify a significant amount of copper without any direct health-damaging effects. In extreme cases, it may even happen that, during this symptom-free period, the copper concentration in the liver increases a hundredfold [110]. However, if the copper concentration of the liver cells exceeds their storage capacity (150–200 mg/kg of wet liver), the cells are damaged, a large amount of copper suddenly enters the bloodstream and enters the red blood cells, resulting in severe hemolysis due to the peroxidation of the membrane lipids [111]. At the same time, the number and quantity of oxidative intermediates entering the blood increases, so part of the hemoglobin is transformed into methemoglobin through oxidation processes, which leads to the development of oxygen transport disorders. Cell organelles are damaged—primarily lysosomes—and the functioning of many vital enzymes stops [112,113].

In animal experiments on domestic hens [90] and on lower organisms [114,115], damage to the genetic material occurred, while some authors drew attention to the possible tumorigenic effect of copper-containing compounds. In the case of chronic copper sulfate poisoning, fetal damage and reproductive disorders may occur [116,117].

### 3.2. Body Weight

The body weights of the chicken embryos were significantly reduced due to the injection of copper sulfate and cadmium sulfate into the eggs [118]; this was also found in our study due to the treatment of Cd, Cu, and Pb by both application techniques. However, compared with the control groups, the reduction of body weight caused by all metals was higher due to the injection application (Cu: 6.90%, Cd: 8.60%, Pb: 9.10%) than in the case of the immersion treatment (Cu: 3.60%, Cd: 6.70%, Pb: 6.90%).

Similarly, the growth of the embryos was significantly reduced during the immersion of mallard eggs into water containing lead nitrate [119]. However, there was no statistical changes described in mallard embryos after injection and immersion when applying Pb and Cd [101].

Cadmium and lead can reduce the growth rate in little blue heron chicks (*Egretta caerulea*) [120].

However, there was no significant differences of body weight in the mallard (*A. platyrhynchos*) embryos due to the immersion treatment with copper and lead [28].

A reduction in embryo weight caused by Cd was also described in laboratory experimental mammals [48,102].

### 3.3. Developmental Aberration

The developmental abnormalities of the mallard embryos (*A. platyrhynchos*) were statistically increased due to the injection of Pb (22.20%) and Cd (25.50%) into the eggs; however, the immersion of the eggs into water containing these metals resulted in a non-significant increase in these alterations (Pb: 16.00%, Cd: 10.40%). During the experiment, retarded growth, open body cavities, and underdevelopment of the blood vessel network were found due to both treatment methods [101].

The ratio of abnormalities was closely similar to our findings in the case of injection of the investigated metals at early (10–20.00%) and late stages of embryonic development (9.38–13.64%). However, there was no aberrations found after the immersion treatment at the early stage, and there was a low incidence of them at the late stage (8.80–12.24%).

Similarly to our results, a blood ring was detected in a Wrolstad medium white turkey embryo on day 10 after incubation [121]; however, it was found with a retarded vascular system on day 2–3 of hatching in our study.

Developmental abnormalities were not found in the mallard (*A. platyrhynchos*) embryos treated with Cu by immersion; however, the Pb induced a significant increase in the anomalies, including blood ring, absence of vascular network, and retarded growth [28].

Twins in avian embryos rarely occur, but are described by several authors in different bird species, e.g., mallard (*A. platyrhynchos*) [122], wedge-tailed shearwater (*Puffinus pacificus*) [123], broiler chicken (*Gallus domesticus*) [124], and ostrich (*Struthio camelus*) [125]. Generally, these developmental abnormalities are detected in the late embryonic stage; however, Siamese twins were found in our study on day 3 of hatching (at the early developmental stage).

Ocular malformations, e.g., anophthalmia, microphthalmia, atrophy, size reduction of eyeball, and underdevelopment, are frequently described in adult domestic fowl and different wild birds [126]. However, the abnormal differentiation of the eyeball was found in chicken embryos at the early stage of development on day 2–3 in our study.

Generally, the beak deformities in birds may develop due to different contaminants, nutritional problems, bacterial or viral diseases, parasites, blunt trauma, or genetic changes.

Based on the scientific literature, beak deformities, e.g., crossed mandibles, upper mandible decurved or upcurved, lower mandible upcurved or decurved, elongation and lateral curvature, can be found in wild birds and domesticated poultry species [127,128].

Crossed-bill deformity was described in the bald eagle (*Haliaeetus leucocephalus*), and the researchers assumed that the causative agents could be contaminants (e.g., heavy metals), among others [129].

Malformations of beak (crossing and pronounced curvature, downward bent, lateral deviation of the top of upper jaw, etc.) were found in different penguin species (emperor penguin [*Aptenodytes forsteri*] and Adélie penguin [*Pygoscelis adeliae*]) in Antarctic ecosystems [130].

Zylberberg et al. [131] stated that a special beak deformity (different degree of lateral abnormality, crossing and gapping of upper and lower bill) called avian keratin disorder, which can appear in all continents in black-capped chickadees (*Poecile atricapillus*), is caused by picornavirus [131]. However, other researchers did not find clear evidence of its bacterial, viral, or fungal origin [132,133,134].

Basically, the researchers hypothesized a genetic background of the malformation of the beak. However, the shortening of the upper beak was found in chicken embryos due to the treatment of copper and cadmium in our study. This does not contradict the previous statement, because Cu and Cd have genotoxic potential, thus they can produce teratogenic alterations.

Leg deformities, e.g., missing toes, duplication or fusion of digit, dislocation or distortion and shortening of bones in the leg, were described in the American kestrel (*Falco sparverius*), peregrine falcon (*F. peregrinus*), merlin (*F. columbarius*), tawny owl (*Strix aluco*), and little penguin (*Eudyptula minor*) [135,136,137,138]. Compared with our results, curvature of the axis of the toes and abnormal position of the leg were found due to the treatment with copper and cadmium.

Similarly, skeletal abnormalities were also described in laboratory mammals (rat, mouse) due to Cd, including the malformation of limbs (sirenomelia, amelia) and other defects, e.g., ossification problem of sternum and ribs and dysplasia of facial bones and limbs [48].

Cadmium can replace the essential metals (e.g., zinc, calcium) in the structural or enzymatic proteins based on their structural similarity and physical/chemical properties, resulting in a deformed “design” of different bones, or abnormal or malformed enzymes, resulting in altered functions. Through its redox activity, it damages the antioxidant system, thus causing oxidative stress, increasing lipid peroxidation, and changing the lipid composition of membranes. Reactive oxygen radicals created by cadmium can lead to decreased DNA synthesis and DNA strand breaks, resulting in abnormal development of cells, thus abnormality of the tissues and organs [139,140].

## 4. Materials and Methods

### 4.1. Materials

#### 4.1.1. Animals

Chicken eggs of Shaver Rusticbro species (*Gallus gallus f. domestica*) were purchased from Goldavis Kft. (Sármellék, Hungary). During the experiment by the injection and immersion methods, 10 eggs at an early development stage and 70 eggs at a late development stage were applied with a treatment of copper sulfate, cadmium sulfate, and lead acetate. The detailed results of the experiment are shown in Table 1.

#### 4.1.2. Test Substances

In the past, metals (e.g., cadmium, mercury, copper) were extensively used as pesticides against the different undesirable pests during plant protecting activity. However, some of them (e.g., copper) are still applied today. Copper compounds are used in micronutrient fertilizers and also as active ingredients in fungicides, which can create opportunities for exposure to wild bird eggs [141].

Furthermore, different heavy metals are found in the environment as natural components, and they can enter the environment due to anthropogenic (industrial, agricultural, traffic) activities. Persistent heavy metals (e.g., mercury, lead, cadmium) found in the environment due to both sources can enter the body of wild mammals and birds. Heavy metals are not biodegradable, and they can be accumulated in living organisms and can be metabolized mostly to more toxic, rarely to less toxic, derivatives by biochemical processes. Thus, due to their environmental polluting and accumulation properties, their enrichment in the environment is highly important from environmental protection aspects [142,143,144,145].

However, due to the previous widespread use, the chemical load caused by metals has by no means disappeared. During the continuous long-term intake of small amounts, they can cause harmful effects (micro-toxicological alterations) or even toxic symptoms due to their cumulative properties (tumor formation, reproduction biological disorders, immune damage). Thus, the investigation of their potential harmful effects in the environment, and on wild animals living in it, is very important to ensure a livable environment [146,147,148].

During both treatment methods (injection and immersion), copper sulfate (Reanal-Ker Ltd., Budapest, Hungary), cadmium sulfate (Reanal-Ker Ltd., Budapest, Hungary), and lead acetate (Reanal-Ker Ltd., Budapest, Hungary) were applied at a dose of 0.01% based on preliminary studies, using their findings on the mortality and the incidence of developmental disorders [149].

An avian physiological saline solution (0.75% sodium chloride) was used to dilute the test substances and for the treatment of the control eggs.

### 4.2. Methods

#### 4.2.1. Treatment

The time of treatment with the chemicals (copper sulfate, cadmium sulfate, lead acetate) was the start of hatching, i.e., day 0, to model the contamination of the egg and the embryo at the beginning of development. The eggs were randomly grouped based on their size and weight and were marked.

##### Treatment Method

The investigated metals were applied to the eggs by two methods, i.e., the injection and the immersion methods.

##### Injection Method

During the injection method, a volume of 0.05–0.10 mL of the test substance can be injected into the air chamber [150,151,152,153,154,155], into the yolk of the egg [156,157], into the white of the egg [158,159], directly under the air chamber [160], onto the surface of the chorioallantoic membrane (CAM), the allantoic duct, the hyoid duct, the embryo, or intravenously [161]. The advantage of the procedure is that the substance to be tested can be delivered in a precisely measured dose to any part of the egg. The disadvantage is that it does not properly model the effect in the environment. By injecting into the air chamber, mechanical damage inside the egg can be avoided, which could affect embryotoxicity. The pre-drilled hole in the lime shell must be sealed with soft paraffin after the treatment.

In our study, a hole was drilled (Forte 300 dental drill, Netdent Trade Ltd., Szeged, Hungary) on the eggshell above the air chamber, and the solution of the investigated metals (0.1 mL of each) was injected into the air chamber by a Nichipet Air micropipette (Nichyro Co., Ltd., Tokyo, Japan). Then, the eggshell was closed by paraffin, and put into the incubator.

##### Immersion Method

Based on the scientific literature, the immersion treatment of the eggs usually means placing them in a liquid corresponding to the hatching temperature for 30 min. Then, after drying, the eggs can be placed in the incubator [151,160,162,163,164,165,166].

If a harmful effect of the chemical is observed in the embryo after the injection application, it is definitely necessary to treat the eggs by immersion.

Use of the immersion treatment is justified because the effect of the substance that actually occurs in the natural environment can be better modeled—primarily in the case of wild bird species—however, it can be mentioned as a disadvantage that it only allows the pesticide to have an indirect effect on the embryo.

In our study, the eggs were immersed in water (37 °C) containing the concentration of the investigated metals for 30 min. After the treatment period, the eggs were dried with filter paper to remove the water, and then they were put into the incubator.

#### 4.2.2. Hatching

The eggs were incubated in a Ragus type table incubator (Vienna, Austria) after a 24 h rest. During hatching, adequate temperature (37–38 °C) and relative humidity (65–75%) was ensured, and the eggs were rotated.

#### 4.2.3. Processing

Processing for investigation of the embryos in the early developmental stage was performed on day 2 and 3 of incubation (preparation of germinal disc), while the investigation of the embryos in the late developmental stage was performed on day 19 of incubation (necropsy, skeleton staining, histopathology).

##### Preparation of Germinal Disc

The eggshell was broken up above the air chamber and the shell membrane and the egg white were removed above the embryo (Figure 5). A filter paper of adequate size was put onto the germinal disc, then the paper was cut around, and removed with the attached embryo and placed into an avian physiological saline solution (0.75% NaCl) at a temperature of 38 °C. The germinal disc was separated from the filter paper disk and floated onto a glass slide. It was painted with 0.1% osmium tetroxide and fixed after removing the unnecessary saline solution. After all this, the slide was fixed with histological glue and covered with a cover plate.

The germinal disc was evaluated using the Hamburger–Hamilton type stages, and vitality, age, status of development, and morphological changes were determined [167].

##### Skeleton Staining

On day 19, the eggs were removed from the incubator and the eggshells were opened by tweezers and scissors, the embryos were investigated, and the following parameters were determined: body weight, number of dead embryos, and type and frequency of macroscopic developmental abnormalities.

To investigate the malformations and disorders of the skeleton, skeleton-stained preparation was prepared using alizarin red dye. The staining procedure was performed by Dawson’s method [168]. For this aim, the feathers, skin, fat tissues, and the eyes were removed, and the body was fixed in ethanol (96%) for 2–3 days for dewatering purposes. Then, it was put into 1% potassium hydroxide solution for up to 2–3 days for maceration. The skeleton was stained by alizarin red dye (1:10,000) dissolved in a potassium hydroxide solution. After that, the body was put into 1:1 mixture of ethanol (96%) and glycerin, and it remained in this solution for 12–19 days. After all this treatment, the stained body was placed into 87% glycerin, in which it could be stored indefinitely. The evaluation of the changes of the skeleton can be performed visually or by using a stereomicroscope.

##### Histopathology

During necropsy, samples were taken from the musculus longissimus dorsi and the liver of three birds of each group (control and metal-treated).

The samples were fixed in 4% formaldehyde solution, and then in paraffin block. After slicing, the ultrathick slices were stained by hematoxylin and eosin (HE), and then they were evaluated under a light microscope [169,170].

#### 4.2.4. Statistical Analysis

All statistical analysis was performed using R statistical data analysis program [171].

The embryo body weight was analyzed by one-way ANOVA. Its assumptions were checked visually by quantile–quantile plots and statistical tests. Normality was checked by a Shapiro–Wilk test, and homogeneity of variances by Levene’s test. These tests approved that both normality and homogeneity can be assumed. The pairwise comparisons were carried out by using Tukey’s HSD test.

The results of the mortality and the developmental abnormality of the embryos were analyzed statistically using Fisher’s exact test [172,173].

## 5. Conclusions

Compounds found in the environment under natural conditions or entering the environment through anthropogenic activity, such as heavy metals, due to their significant cumulative property as an environmental burden pose a high degree of risk to living organisms. Metals from natural sources (fossil materials, volcanic rock, ores) can repeatedly enter the different environmental elements (water, soil, air) during industrial use or even natural erosion. During mining activity (dust, slag, soot), through processing industries (smoke, solid particles, combustion products), and during the production of commercial products (e.g., pesticides, paints, medicines), metals released through industrial emissions (e.g., sewage) can pollute vegetation. They can enter surface waters, and, by penetrating the waterproof layers, they can enter the water network, and they can also cause soil contamination. Human origin (municipal, industrial) emission within the same time interval can be orders of magnitude higher than the element content from natural sources. These pollutants can continuously worsen the state of the environment. However, it is important to know that different territories, e.g., agricultural areas, are a source of food and hiding and breeding places for wild birds, so these compounds can affect not only adult birds but also embryos developing in the eggs, impairing their viability, or even causing morphologically observable abnormalities. The literary sources confirm that the avian embryo can be used in first-line embryotoxicity and teratology studies, as it reacts to the damaging effects of various chemical and physical agents with great sensitivity. However, there are/may be differences in the sensitivity of wild bird species to different chemicals; furthermore, the toxic effects may depend on species, age, health status, and exposure parameters (dose, duration, frequency). Morphological and functional changes in the embryogenesis of birds are similar to the embryonic development of mammals in many ways, which gives the possibility for extrapolation.

Certain metals, e.g., cadmium, copper, and lead, based on our investigations, can penetrate through the eggshell and membranes into the germinal disc or embryo at early or late development stages, leading to decreased body weight, retarded growth rate, developmental malformation of offspring, abnormalities, or even death. Unfortunately, these effects can endanger the reproduction, or can even induce the extinction of species. Thus, the evaluation and knowledge of the results raised from avian teratological studies of environmental polluting heavy metals can greatly help to protect living organisms in the environment as best as possible.

The test series can be supplemented by measuring the mass of organs removed for histological processing (heart, liver) and other organs, with histopathological examinations covering additional organs in addition to the long neck muscle and the liver, along with FTIR (Fourier transform infrared spectroscopy) and FT-Raman (Fourier transform Raman) spectroscopic processing of histological sampling from liver and brain, and beak, claw, and feather sampling to detect changes caused by heavy metals, with the biochemical and dynamic analysis of the chemical substances used, and their metabolites, in the embryonic age from blood samples obtained from the umbilical artery using a glass capillary [174,175], by examining some blood plasma parameters [glucose, calcium, magnesium, inorganic phosphate, AST (aspartate aminotransferase), ALT (alanine amine transferase), ALP (alkaline phosphatase), LDH (lactate dehydrogenase), pChE (pseudocholinesterase), total protein, and albumin].

Due to the increased sensitivity of wild bird species, we recommend conducting the tests on waterfowl (wild ducks) and seed-eating species (pheasants, Japanese quail). The experiments can be supplemented with hatchability and post-embryonic tests, which can provide additional information regarding the individual and combined toxicity of the substances used in biological systems.

## Figures and Tables

**Figure 1 ijms-25-10662-f001:**
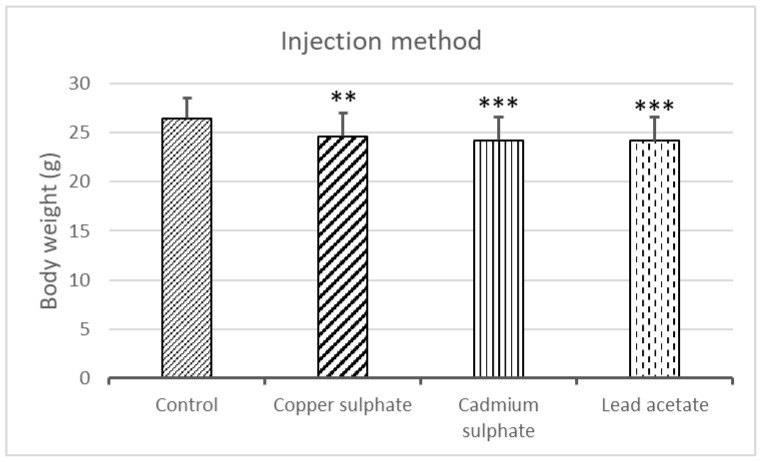
Evolution of the embryonic body weight due to injection treatment. [** Significant difference from the control (*p* = 0.0021); *** significant difference from the control (Cd: *p* = 0.0003; Pb: *p* = 0.0004)].

**Figure 2 ijms-25-10662-f002:**
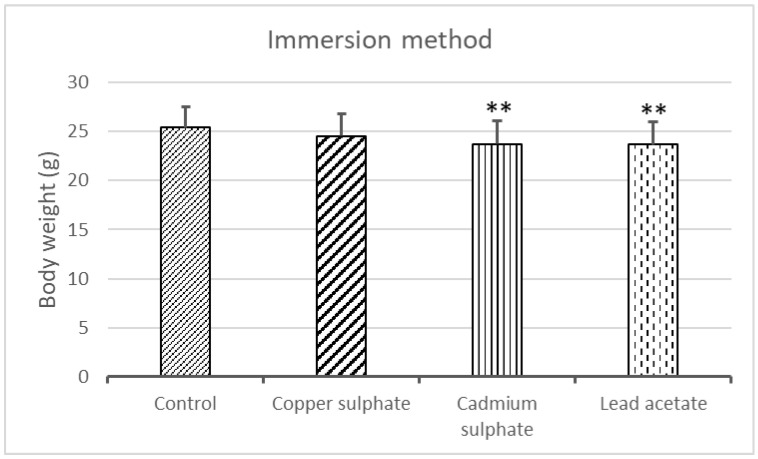
Evolution of the embryonic body weight due to immersion treatment. [** Significant difference from the control (Cd: *p* = 0.0025; Pb: *p* = 0.0013)].

**Figure 3 ijms-25-10662-f003:**
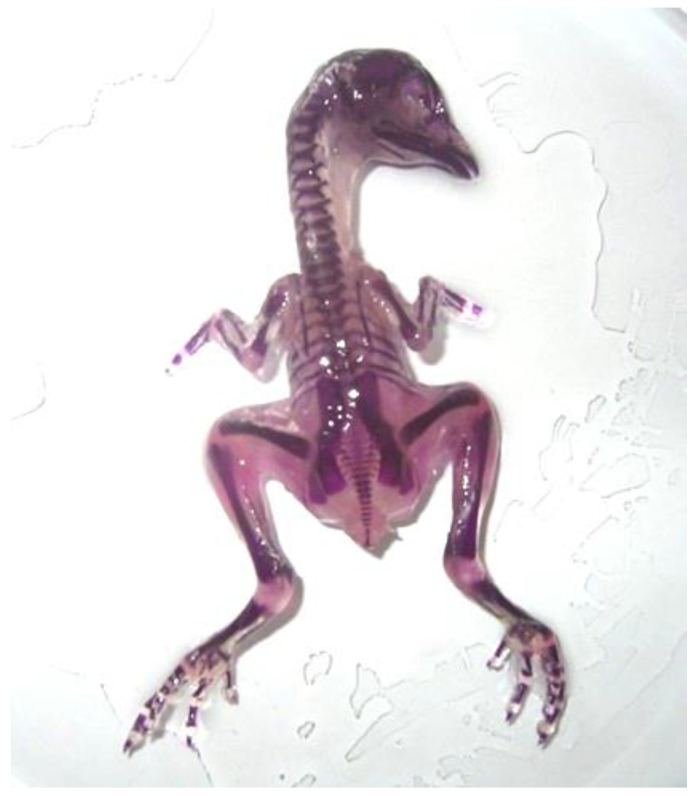
Skeleton-stained control embryo without deformities on day 19 of hatching.

**Figure 4 ijms-25-10662-f004:**
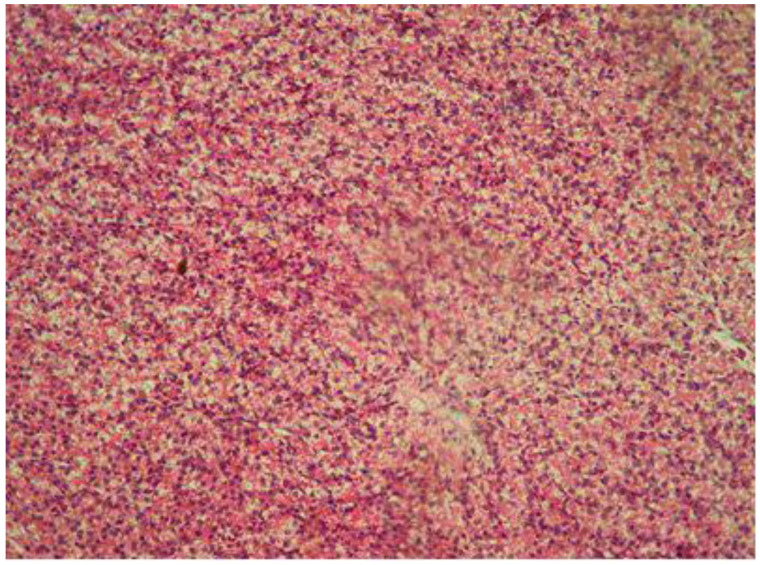
Liver tissue from the control embryo (H-E staining, 400× magnification).

**Figure 5 ijms-25-10662-f005:**
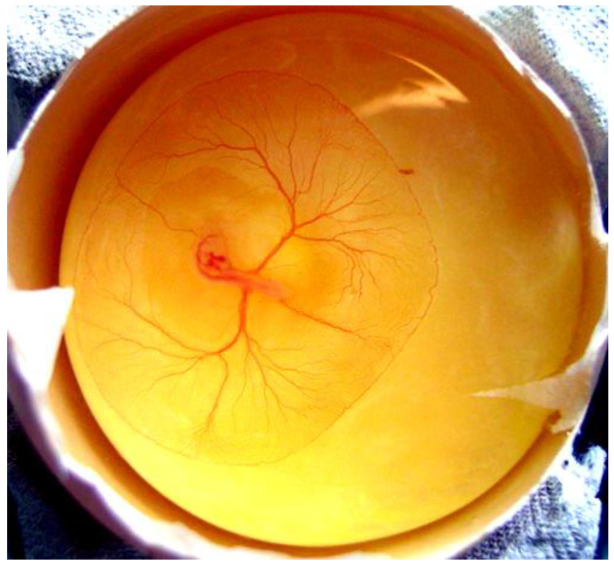
Chicken embryo on day 3 of incubation (control embryo).

**Table 1 ijms-25-10662-t001:** Study design.

Group	No. of Eggs			
	Early Stage		Late Stage (Day 19)	
	Day 2	Day 3	Macroscopic Processing, Histopathology	Skeleton Staining
Control	5	5	50	20
Copper sulfate	5	5	50	20
Cadmium sulfate	5	5	50	20
Lead acetate	5	5	50	20

## Data Availability

The data used to support the findings of this study can be made available by the corresponding author upon request.

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
