# Peer review of "Potential Teratogenicity Effects of Metals on Avian Embryos"

_ijms, 2024, doi:10.3390/ijms251910662_

Round 1

Reviewer 1 Report

Comments and Suggestions for Authors

Potential Teratogenicity Effects of Metals on Avian Embryos

This paper investigates the potential teratogenic effects of heavy metals (cadmium, copper, and lead) on avian embryos. By using chicken embryos as a model, the study examines the impacts of these metals at both early and late stages of embryonic development. Overall, the research offers an in-depth analysis of the adverse effects of heavy metal exposure on chicken embryo development, providing valuable scientific insights into environmental pollution and its impact on wildlife.

However, there are several points that require further clarification and refinement. First, under typical environmental conditions, it is unlikely that chicken embryos would be exposed to such high concentrations of heavy metals as those simulated in this study. While heavy metal pollution is a widespread concern, concentrations in the natural environment are usually lower than those used in the experiment. Therefore, I would like to know how the authors determined the concentrations of heavy metals used in their tests. Are these concentrations reflective of real-world exposure levels, and do they meet the experimental requirements?

Additionally, I have concerns regarding the method of injecting heavy metals directly into the eggs. Is there a scientific basis or precedent for using this technique? A clearer explanation or rationale for choosing this method would strengthen the study's credibility.

For the comparison between the injection method and the immersion method, I recommend a more detailed analysis of the differences between the two approaches, as well as an exploration of the potential reasons behind any variations in results.

Finally, in the assessment of early and late-stage embryonic development, I suggest including some physiological and biochemical indicators, such as body weight and organ indices. This would provide a more comprehensive evaluation of the toxic effects of the heavy metals and help to better quantify their impact.

Author Response

Please see the attachment,

Reviewer 2 Report

Comments and Suggestions for Authors

Major Comments:

1. The manuscript lacks a comprehensive justification for the selection of the doses and the specific metals (Cd, Cu, Pb) used in the study. The authors should provide a more detailed rationale, possibly supported by previous studies, to explain why these specific concentrations and metals were chosen and how they are relevant to environmental exposure scenarios.

2. The description of the statistical methods used is insufficiently detailed. The manuscript should include more information on the statistical tests employed, including any assumptions tested, post-hoc analyses conducted, and effect sizes. This would provide readers with a clearer understanding of the robustness and reliability of the findings.

3. While the study presents significant findings regarding embryonic mortality and developmental abnormalities, the discussion lacks depth in interpreting these results in the context of existing literature. The authors should expand on how their findings compare to previous studies on the teratogenic effects of metals and what new insights their study adds to the field.

4. The manuscript would benefit from a more detailed exploration of the potential biological mechanisms underlying the observed toxic effects of the metals on avian embryos. This could involve discussing the known pathways of metal toxicity at the cellular or molecular level and how they might relate to the observed developmental abnormalities.

5. The study should better address the environmental relevance of the findings. This includes a discussion on the real-world implications of the observed teratogenic effects for wildlife conservation and environmental policy. The authors should also suggest potential mitigation strategies or further research needed to explore these effects in other species or under different environmental conditions.

Minor Comments:

1. There are several typographical errors throughout the manuscript. For instance, on page 4, "the white of egg were removed" should be corrected to "the white of the egg was removed." A thorough proofread is recommended to correct such errors.

2. The figures included in the manuscript, particularly those depicting the histological findings, are of low resolution and lack detailed legends. Higher quality images with comprehensive legends explaining all relevant features should be provided.

3. The manuscript uses varying terms to describe the same concepts (e.g., "embryo mortality" and "fetal mortality"). Consistency in terminology should be maintained throughout to avoid confusion.

4. Several references in the manuscript do not follow the journal's formatting guidelines, such as missing DOIs or improperly formatted journal names. The authors should carefully check the reference list to ensure compliance with the required format.

5. The description of the experimental setup, particularly the immersion and injection methods, could be made clearer. Adding a schematic diagram or a more detailed explanation of the procedure would help readers better understand the methods used.

Round 2

Reviewer 2 Report

Comments and Suggestions for Authors

No more comments